# Collaborative research to support urban agriculture in the face of change: The case of the Sumida watercress farm on Oʻahu

Jennifer L. Engels[1]☯*, Sheree Watson[2]☯, Henrietta Dulai[3,4]☯, Kimberly M. Burnett[5]☯, Christopher A. Wada[5]☯, ʻAnoʻilani Aga[6]‡, Nathan DeMaagd[5,7]‡, John McHugh[8]‡, Barbara Sumida[9]†‡, Leah L. Bremer[4,5]☯

1 Hawaiʻi Institute of Geophysics and Planetology, University of Hawaiʻi at Mānoa, Honolulu, Hawaii, United States of America, 2 Pacific Biosciences Research Center, University of Hawaiʻi at Mānoa, Honolulu, Hawaii, United States of America, 3 Department of Earth Sciences, University of Hawaiʻi at Mānoa, Honolulu, Hawaii, United States of America, 4 Water Resource Research Center, University of Hawaiʻi at Mānoa, Honolulu, Hawaii, United States of America, 5 University of Hawaiʻi Economic Research Organization, University of Hawaiʻi at Mānoa, Honolulu, Hawaii, United States of America, 6 Institute of Hawaiian Language Research and Translation, University of Hawaiʻi at Mānoa, Honolulu, Hawaii, United States of America, 7 Department of Economics, University of Hawaiʻi at Mānoa, Honolulu, Hawaii, United States of America, 8 Pesticides Branch, Hawaiʻi Department of Agriculture, Honolulu, Hawaii, United States of America, 9 Sumida Farm, Aiea, Hawaii, United States of America

☯ These authors contributed equally to this work.
† Deceased.
‡ These authors also contributed equally to this work.
* engels@hawaii.edu

**Data Availability Statement:** All relevant data are within the manuscript and its Supporting Information files.

## Abstract

As urban areas expand around the world, there are growing efforts to restore and protect natural and agricultural systems for the multitude of ecosystem services they provide to urban communities. This study presents a researcher-farmer collaboration in a highly urbanized area of Oʻahu focused on understanding the historical and current challenges and opportunities faced by a culturally and socially valued spring-dependent urban farm, Sumida Farm, which produces the majority of the state of Hawaiʻi's watercress. We conducted a long-term trend analysis (25 years) of factors identified by the farmers to be important historical drivers of crop yield, including groundwater pumping, pest outbreaks, temperature, Oceanic Niño Index, and precipitation. We combined this analysis with a year of intensive spring water sampling on the farm to evaluate nutrient and contaminant composition and flow to understand water-related stressors, as well as evaluate the potential of the farm to provide nutrient retention services. We found negative correlations between historical crop yields and increases in the Oceanic Niño Index, temperature thresholds, and pest outbreaks. Despite the surrounding urbanization, we found on-farm water quality to be very high, and microbial analyses revealed an abundance of denitrifiers (*nirS* gene) suggesting that the farm provides a nutrient retention service to downstream systems. Finally, we found that socio-cultural values including heritage value, aesthetic value, and educational value are increasingly important for the Sumida family and surrounding community. These socio-cultural benefits alongside highly valued local food production and nutrient retention services are essential for continued community and political support. Collectively, our study

**Funding:** Jennifer Engels received funds to conduct this work from the U.S.A. National Science Foundation Integrative and Collaborative Education and Research Division; Award #: 1645515; Award Name: Collaborative Research: Active Societal Participation in Research and Education. These funders required that the research team be a Mobile Working Group that included community partners. Additional support came from the Hawaiʻi EPSCoR Program; U.S.A. National Science Foundation's Research Infrastructure Improvement (RII) Track-1: ʻIke Wai: Securing Hawaiʻi's Water Future Award # OIA-1557349. The funders had no role in data collection and analysis, decision to publish, or preparation of the manuscript.

**Competing interests:** The authors have declared that no competing interests exist.

demonstrates that challenges facing urban agricultural systems shift through time, and that recognition of the beyond crop-yield benefits of these systems to urban communities is essential to their long-term survival.

## Introduction

Concerns about the well-being of growing urban populations globally has led to increasing interest in urban ecosystems and ecosystem services, including provisioning (e.g. crop yields), regulating (e.g. nutrient retention, storm water regulation), supporting (e.g. nutrient cycling) and cultural (e.g. mental health benefits, sense of place) services [1–3]. Urban ecosystems include natural, novel (e.g. constructed wetlands and green roofs), and urban agricultural systems such as community gardens [4–9]. Within the broader context of urban ecosystem protection, there is growing interest in protecting remaining pockets of agriculture in rapidly urbanizing areas, in part because of links to human well-being through local food production, aesthetic value, and other ecosystem services [10]. In particular, the potential for urban wetland agriculture to provide many of the regulating ecosystem services of natural wetlands, while also providing local food and other services, is becoming more established [11, 12].

In a review of urban ecosystem services, McPhearson et al. (2014: 502) [13] state: "designing, planning, and managing complex urban systems for human health and well-being require urban ecosystems to be resilient to systemic change, and to be managed sustainably to provide critical ecosystem services reliably over time." This requires greater attention to the influence of changing environmental, social, and political conditions on urban ecosystems, as well as learning from systems that have effectively persisted, adapted and thrived in the face of change. Particularly in the case of agricultural systems, the way that these systems are valued (i.e. for crop production only or for a diversity of services), is a critical part of adapting to changing conditions. Previous research has demonstrated that, in some areas, small farms are rarely economically viable by crop production alone and that many successful small farms rely on grant and other revenue streams based on diverse benefits (e.g. aesthetic value; farm experience) provided by these systems [14–16].

We present a case study from a spring-dependent watercress farm (Sumida Farm) in the Pearl Harbor aquifer on the island of Oʻahu to illustrate the historical and current challenges faced by urban agricultural systems, as well as the multiple, beyond-crop yield benefits they provide. This farm is one of the last pockets of agriculture in one of the most highly urbanized areas in Hawaiʻi, and is reliant upon natural spring discharge (Kalauao Spring) from the most heavily utilized aquifer in the state [17]. Our aim is to improve understanding of the factors contributing to watercress yield, as well as the other benefits that have led to Sumida Farm's persistence over time, with the hope of contributing to its ability to operate in the future. In doing so, we shed light more broadly on the potential futures of spring-dependent urban agricultural systems that are highly valued for a suite of ecosystem services [1]. Specifically, we utilized mixed methods (Table 1) including trend analysis, field water quality and microbial sampling, and semi-structured interviews to address the following research questions:

1. What factors may have influenced watercress yields over the past 25 years?

2. What is the current quality and quantity of spring water on the farm and how does it relate to the farm's ability to continue to produce watercress?

3. What additional socio-cultural benefits are provided by the farm to the Sumida family and surrounding community?

**Table 1. Quantitative and qualitative factors hypothesized to influence ecosystem services provided by Sumida Farm, main effects, and assessment methods.**

| Factor | Spring flow | Climate | Urbanization | Owners' desire to continue farming | Community and State value of local farms |
|---|---|---|---|---|---|
| **Main effect** | Water conditions conducive to provisioning (crop growth), and regulating and supporting (nitrogen fixation, nutrient cycling, pest resistance) ecosystem services [1] | | Pollution source | Cultural ecosystem services including continuation of heritage farm, aesthetic and educational benefits, sense of place, social relations (community visits), agritourism [1] | |
| **Assessment method** | groundwater pumping data, interviews with farm owners, literature | 25-year datasets for air temperature, precipitation, Oceanic Niño index; 1-year water quality study for salinity | 1-year water quality study (legacy pesticides, nutrients, pharmaceuticals) | Interviews with Sumidas | Hawaiian language newspaper translations, interviews with stakeholders, analysis of local and national press |

## Methods

### Study site

Sumida Farm is located within the Pearl Harbor aquifer in the ʻEwa moku (district) of the island of Oʻahu (Fig 1), a region of rapid historical and ongoing change. Prior to Western contact in 1778, this region was highly valued for the abundance of water and agricultural systems, most notably wetland taro (loʻi kalo) and colluvial agroforestry systems [18, 19]. The area remained dominated by taro farming through the 1800s, shifting towards diversified agricultural systems, rice, banana, and other staples by the early 1900s, followed by widespread sugar production through the 1950s [20–23]. The decline of sugar in Hawaiʻi in the 1960s and conversion of plantations into urban development led to the rapid transition of the area into a dominantly residential, commercial, and militarized landscape, with a few remaining pockets of agriculture and conservation land in lowland areas, leading to high loss of wetland areas [24].

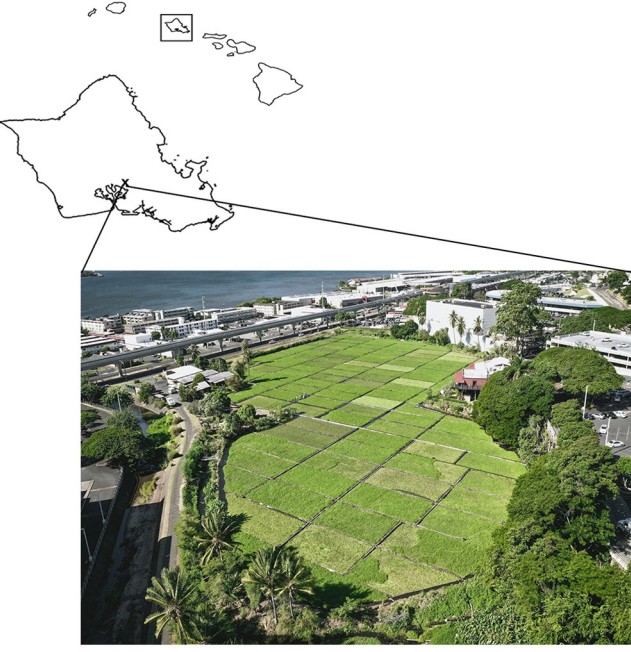

**Fig 1. Map of Hawaiian Islands including the location of Sumida Farm on Oʻahu (X).** Call-out shows aerial image looking southwest across Sumida Farm (published under a CC BY license, with permission from photographer Corey Rothwell, original copyright 2018). Grid layout consists of individual watercress plots. Surrounding urban zones include the Pearlridge Mall and the Honolulu Rail Transit system. Pearl Harbor appears on the horizon.

The four major spring complexes in the Pearl Harbor region have all declined in flow by approximately 50% since 1880 [17]. One of these springs, Kalauao Spring (34,000 m$^3$/d [25]), supports the Sumida Farm, a four hectare multi-generational family farm founded in 1928 that provides 70% of the watercress in Hawai'i, and relies 100% on spring flow (Fig 1). This is one of the last multi-generational family farms in this now highly urbanized area.

Springs that provide water to the farm discharge along the inland margin of the caprock covering the shoreline of the harbor. Water outcrops from discrete springs from orifices where basalt is exposed, and as diffuse seeps where the caprock is thin or erosion has exposed basalt [23]. The upland margin of the farm is defined by a break in slope in the land surface where multiple individual springs are identified. The middle springs discharge through a thin layer of caprock on the flat farmed area and have more voluminous discharge but are fewer in number.

A sprinkler system installed in the 1980s cools the crop and mitigates pest problems by recirculating spring water [26]. The Sumidas lease the land from the State's largest private landowner, and operate the farm with eleven full time employees who hand plant, harvest, and wash 4–5 tons of watercress per week. While the farm continues to produce a substantial amount of watercress, over the past few decades yields have declined by 30–40% (Fig 2) and the Sumidas are concerned about threats from pests, increasing salinization of springs, and pollution from surrounding urban development and poor wastewater management. The Sumidas hypothesize that lower yields and diseases such as watercress "rot" (loss of watercress due to parasites experienced the last three summers) in the hotter summer months may be partially due to decreased spring flow and associated increases in salinity.

## Long-term analysis of factors influencing crop yields

In order to understand factors potentially contributing to observed declines in watercress yields at Sumida Farm (Table 1), we began by digitizing monthly hand-written records of watercress harvest (measured in bundles) kept by the Sumida family since January 1994 (Fig 2). Based on recommendations from prior work [26, 27] as well as conversations with the Sumida family, we combined these records of watercress yield with the following monthly data, which were hypothesized by the family and project team to affect watercress yields (data source in parentheses):

Groundwater pumping (Roy Hardy, personal communication, 2018)

Air Temperature (National Centers for Environmental Information, 2019) [28]

Precipitation (National Centers for Environmental Information, 2019) [28]

Oceanic Niño Index (ONI) (National Weather Service Climate Prediction Center, 2019) [29]

Presence of aster yellows disease (John McHugh personal communication, 2018)

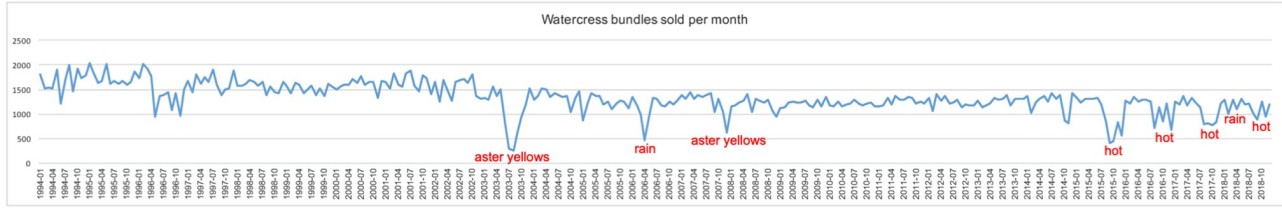

**Fig 2. Monthly harvest of watercress in "bundles" from 1994–2018.** Reasons hypothesized by the Sumida family to have caused lower watercress yields are shown in text under the bundles sold per month graph.

To assess the relationship between high air temperature and watercress production, we considered the minimum, average, and maximum temperatures in each month between 1994–2019. We then created dummy variables indicating high minimum and high average temperatures. We excluded a maximum temperature threshold because a single hot day in a given month could be an anomaly, whereas having a high temperature even on the least hot day of the month and/or high average temperature throughout the month implies that the whole month was hot, which is more likely to affect watercress production. We defined high minimum temperature as any observation equal to or exceeding 24 ˚C, and high mean temperature as any observation equal to or exceeding 30.5 ˚C.

Because the distance from Sumida Farm to each of the groundwater wells in our study area (S1 Fig) varied and pumping from more distant wells was likely to have less influence on spring discharge and watercress production, we estimated the relationship between pumping and watercress yield using the inverse distance-weighted sum of all pumping in the Waimalu groundwater management unit, where Sumida Farm is located in central Oʻahu.

A simple ordinary least squares model was used with the time series data to estimate the relationship between crop harvest and the variables listed above. As a robustness check, we tested models with and without seasonal controls. We also tested the significance of various lags in the explanatory variables (e.g. we compared one month's harvest to the pumping data from up to 12 months prior). We did not include the temperature variables in a regression alongside ONI because ONI and temperature are related; doing so would potentially create a multicollinearity problem.

## Analysis of current spring water quality

**Geochemical and pollutant sampling and analysis methods.**   In order to examine overall water quality within the springs and farm, as well as identify potential anthropogenic influences on the water source, we examined selected springs at Sumida Farm for legacy pesticides, nutrients, and stable isotopes of nitrate to identify agricultural influences (Table 1 and Fig 3). In addition, we used pharmaceuticals as tracers to rule out any wastewater leaks in the water from upstream urban development. We collected spring water from six major springs on the farm in the dry (Sep 2018) and wet (Feb 2019) periods. Water was collected before its discharge to the surface to capture water chemical parameters typical for the aquifer. A push point sampler was used to withdraw water with a peristaltic pump from 0.3–0.5 m in the subsurface. Salinity, temperature, and dissolved oxygen levels were measured in-situ in a flow-through vessel with a YSI multiparameter probe (model YSI Elite Pro 30). Radon was collected in 250 mL glass bottles without head space and analyzed using a RAD-H2O instrument (Durridge, Inc.). Radon measurements were decay corrected to the time of sample collection. Water for oxygen and hydrogen stable isotopes of water analysis was collected in 20 mL glass vials. Water samples for dissolved inorganic and total nutrient analysis, $\delta^{15}N$ and $\delta^{18}O$ of nitrate were collected in acid-washed HDPE bottles, water was filtered using a 0.45 μm capsule filter and kept refrigerated and frozen, respectively, until analysis. Pesticides and pharmaceuticals were filtered using a 0.45 μm capsule filter, collected in 40 mL amber vials and kept refrigerated until analysis. Stable isotopes of water were analyzed at the Biogeochemical Stable Isotope Laboratory at the University of Hawaiʻi (UH), nutrients were analyzed at the School of Ocean and Earth Science and Technology (SOEST) Laboratory for Analytical Biogeochemistry at UH, $\delta^{15}N$ and $\delta^{18}O$ of nitrate were analyzed at the UC Davis Stable Isotope Facility, pesticides (atrazine, glyphosate, DDT +DDE) and pharmaceuticals (carbamazepine, caffeine, and ethynylestradiol) were analyzed using ELISA methods at UH.

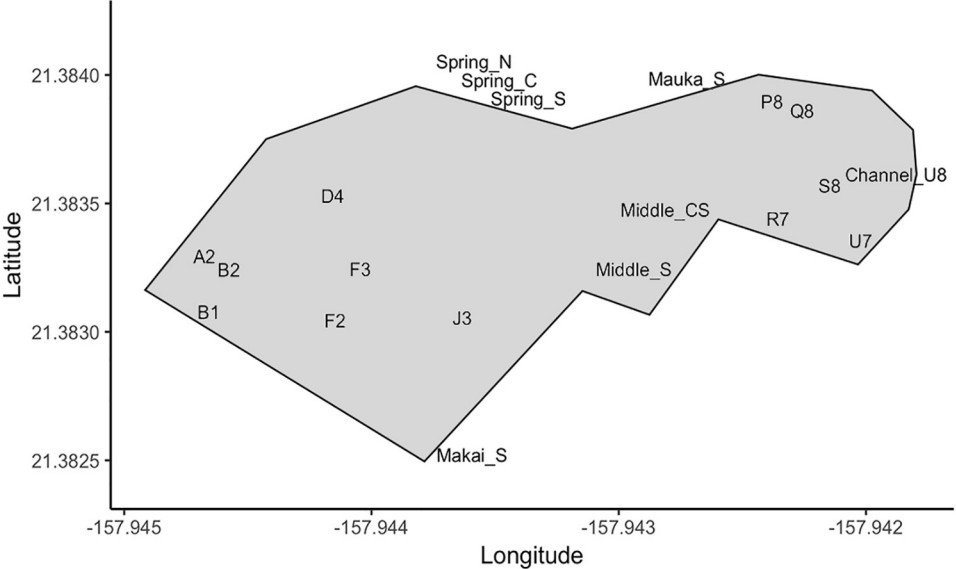

**Fig 3. Map of Sumida Farm including springs and watercress plots sampled monthly and/or bi-annually throughout the year (2018–2019).** Springs are indicated by Name_S, and specific watercress plots are designated by a letter and number.

**Microbial survey for nitrogen (N) cyclers.** Microbial community N cycling was evaluated by collecting water samples monthly from Sumida Farm from the surface at the mauka ("mountain", i.e. upstream) spring, middle spring, and makai ("ocean", i.e. downstream) spring (sump pump) from May 2018 to April 2019 (Fig 3). In addition, we collected water samples from plots in the farm in various stages of health, and growth stages from plots in the upper half of the farm (P8, Q8, S8, R7, U7, U8; Fig 3) and from the lower half (A2, B2, B1, F2, F3, J3, D4; Fig 3) from June 2018 –February 2019. One liter of water was collected following 3 sample rinses in a sterile Nalgene bottle, stored on ice, and returned to the lab within two hours for filtering. Water was sterile filtered in two– 500 mL replicates on 0.8 μm and 0.2 μm polycarbonate membrane filters (Whatman, NY) and stored at -80˚C until DNA extraction. Approximately 250 mL of filtered water was saved in acid washed polycarbonate bottles, and stored at -20˚C for nutrient analysis at the SOEST laboratory for analytical biogeochemistry. Water samples were analyzed for dissolved inorganic nutrients including: ammonium, nitrate+nitrite, phosphate, silicate, and total nitrogen and phosphorus.

DNA was extracted from frozen filters using the DNeasy Power Water Kit (QIAGEN, MD) according to the manufacturer's instructions. DNA extracts were stored at -20˚C until quantitative PCR (qPCR) was performed. Primers utilized for amplification in this study are listed in the Supplementary Information (S1 Table) for genes encoding bacterial *16S*, nitrogen fixation (*nifH*), denitrification (*nirS*), and bacterial ammonia oxidation (*amoA*). Quantitative PCR was performed on an Applied Biosystems StepOnePlus real-time machine (ThermoFisher, MA). All reactions were performed in a 20 μL volume reaction mixture containing 10 μL of PerfeCTa SYBR Green FastMix (Quantabio, MA), 0.5 μM of each primer, ~20 ng of total DNA, and molecular grade PCR water. Thermocycling qPCR conditions for Bacterial *16S* amplification included 94˚C for 10 min, 35 cycles of 94˚C for 60s, 59˚C for 60s, and 72˚C for 30s. Conditions for Nitrogenase (*nifH*) included 94˚C for 10min, 30 cycles of 94˚C for 60s, 55˚C for

60s, and 72˚C for 60s. Amplification for Nitrite reductase (*nirS*) included 95˚C for 30s, 15 cycles of 95˚C for 15s, 66˚C for 20s, and 72˚C for 20s, followed by 30 cycles of 95˚C for 15s, 60˚C for 20s, and 72˚C for 20s. Finally, conditions for Anammox (*amoA*) included 95˚C for 10min, and 40 cycles of 95˚C for 10s, 55˚C for 30s, and 72˚C for 60s.

Assay plates were covered with adhesive film and spun at 1500 rpm for 2 cycles of 40 seconds to remove bubbles from sample wells. Standards were created using gBlock Gene Fragments, double stranded DNA fragments (Integrated DNA Technologies, IA) created by aligning primers with genes obtained from representative sequenced microbial genomes containing the gene of interest. Stock solutions of standards were prepared by calculating concentrations for $10^9$ copies ml$^{-1}$ and serial dilutions were performed from $10^8$ to $10^2$ for *16S*, *nifH*, *nirS*, and *amoA* assays. All sample measurements were done in triplicate, and negative controls (blanks) were included in all assays. Specificity was checked with agarose gel electrophoresis and melting curve analysis at the end of each qPCR assay.

**Socio-cultural values of Sumida Farm.** To better understand the historical and current social values of Sumida Farm and the Kalauao Spring beyond watercress yield alone, we combined archival analysis of historical newspapers with semi-structured interviews of the current farm managers (the Sumida family) as well as the State water regulator. We obtained University of Hawai'i at Mānoa Institutional Review Board (IRB) approval for this study and followed standard protocols for prior informed consent. We also reviewed local and national press about the farm that related to its role in the community [30–39].

As with other springs in the Pearl Harbor region, the Kalauao Spring has a long history far beyond the 90+ years of Sumida Farm. One of the most detailed sources for information on springs and places in Hawai'i prior to 1920 (just before the Sumidas acquired the lease) is the Hawaiian language newspapers. From 1834 to 1948 over 125,000 pages of Hawaiian language print were published in more than one hundred independent newspapers [40]. These newspapers are a useful archive of knowledge, opinion, and historical progress covering the period when Hawai'i moved through kingdom, constitutional monarchy, republic, and territory [41]. By the mid-1800s, newspapers encouraged their readers to submit content for the papers, which included detailed reports on weather and volcanic activity, as well as other descriptions and narratives that often showcased the importance of observing the natural world in Hawaiian culture [42]. This newspaper archive includes more than a million typescript pages of text—the largest native-language collection in the Western Hemisphere, yet only ~2–3% of that archive has been translated and utilized in modern research [40, 42]. The Institute of Hawaiian Language Research and Translation (IHLRT) at the University of Hawai'i was established to begin the process of translating the Hawaiian newspapers, and collaborates broadly to utilize newspapers to inform and shape place-based research. Several of our research team members are IHLRT staff and students who conducted an extensive place name search around Kalauao Spring to better understand the way that the spring and surrounding areas were used and valued in the past (Table 1). As with other areas in Hawai'i, the historical value often informs the current cultural value [43], so this research both helped to inform our research and also was of great value to the Sumida family and surrounding community.

Interviews and informal conversations with the Sumida family focused on the history of the farm, key challenges and opportunities, as well as perceptions of the multiple ways that the farm is valued by the family and broader community. Our interview with the head of the groundwater division of the Hawai'i State Commission on Water Resources Management (CWRM; the water regulator) focused on understanding how historical and current water allocations and the state's water code apply to spring-fed agriculture today (Table 1).

## Results

### Results of long-term analysis of factors influencing crop yields

Between 1994–1998, monthly watercress yields averaged over 1600 bundles, but farm output has slowly and continuously declined since, averaging less than 1100 bundles per month over the period 2015–2019 (Fig 2). With the exception of pumping, all relationships (high minimum temperature, high average temperature, precipitation, and aster yellows) had the expected correlation with watercress yields. While the regression of bundles against pumping suggested a significant positive effect (additional 1 million gallons per day (mgd) of pumping was associated with an increase in monthly harvest of 320 bundles), the other variables tested showed negative and statistically significant relationships with watercress harvest (Table 2).

A high minimum temperature was associated with a loss of 200 bundles compared to the average, and was significant at the 0.1% level, while a high average temperature was associated with a loss of 182 bundles compared to the average, significant at the 0.1% level. The significance remained even after using a restricted cubic spline on harvest month to account for unobserved seasonality in climate and other factors affecting harvest. High maximum temperatures were not statistically significant. This was also expected, as watercress yields are likely more affected by a long stretch of heat rather than a day or two of high temperatures. Precipitation had a weak negative relationship with harvest and was mostly driven by outliers. This insignificant result may be because Sumida Farm does not rely on rainfall for irrigation, but rather the natural spring discharge, which is also used in a sprinkler system for pest control. As expected, the presence of the aster yellows disease had a very significant effect on yield. Affected months resulted in a harvest with 725 fewer bundles, on average. Finally, we found that a 1-unit increase in the ONI resulted in 62 fewer bundles on average.

### Results of analysis of current spring water quality

**Geochemical signatures and pollutants.** The farm area receives groundwater from discharge of >10 easily identifiable springs. Spring water quality was assessed in six major springs. A summary of chemical and water quality data is provided in the supplementary information (S2 Table). Water quality parameters in all four sampled springs at the mauka boundary of the farm (Fig 3) varied by less than ~10% of their average suggesting a common water source. The two springs in the middle of the farm were different from the mauka springs

**Table 2. Relationships between watercress bundles and pumping, climate, and pest variables.**

|  | (1) | (2) | (3) | (4) | (5) | (6) |
|---|---|---|---|---|---|---|
| Pumping | 320.29*** (9.571) |  |  |  |  |  |
| High minimum temperature |  | -200.39*** (-3.701) |  |  |  |  |
| High average temperature |  |  | -182.73*** (-3.356) |  |  |  |
| Precipitation |  |  |  | -0.8493* (-2.519) |  |  |
| Aster yellows |  |  |  |  | -725.64*** (-6.762) |  |
| Oceanic Niño Index |  |  |  |  |  | -62.74** (-3.036) |
| Intercept | 534.33 (6.672) | 1311.35 (70.187) | 1309.25 (69.689) | 1316.12 (62.23) | 1304.44 (78.962) | 1282.63 (72.462) |
| Observations | 211 | 211 | 211 | 211 | 211 | 211 |

$t$ statistics in parentheses.

*$p<0.05$,

**$p<0.01$,

***$p<0.001$.

in terms of water chemistry, temperature, salinity, and nutrient content. Salinity was three times higher in the middle springs, 1.23+/-0.13 vs 0.34+/-0.04 than in the mauka springs. Total dissolved nitrogen was 29.0+/-0.4 and 34.9+/-2.1 uM, nitrate plus nitrite 21.1+/-0.8 and 24.8 +/-1.5 uM, total dissolved phosphorus was 1.71+/-0.11 and 2.82+/-0.05 uM, dissolved phosphate 1.32+/-0.11 and 2.48+/-0.06 uM, and finally silicate was 847+/-3 and 920+/-7 uM for middle and mauka springs, respectively. Ammonium was below the detection limit in all springs. $\delta^{15}N$ and $\delta^{18}O$ of nitrate was comparable at the different springs, with averages 5.7 +/-0.4 ‰ and 4.6+/-0.3 ‰, respectively. Radon showed some variability (208+/-60 dpm/L) but was comparable in all springs and did not show seasonal changes.

Pharmaceuticals ethynylestradiol (EE2), carbamazepine and caffeine were below detection limits. The legacy insecticide DDT+DDE not currently used but still persisting in the aquifers throughout the island was found in all of the tested springs with no difference between the upstream and middle springs (2.2+/-0.3 ng/mL). The herbicide atrazine was below the detection limit (0.05 ng/mL) in all samples.

Oxygen and hydrogen stable isotopes of water measured in the winter exactly match the local precipitation in the wet season [44], suggesting that rain locally recharged into the aquifer feeds the springs. Unlike precipitation which exhibited seasonality in the isotope composition, springs had the same values in both seasons suggesting that recharge is dominated by wet season precipitation.

**Microbial N cycling.** The largest concentrations of dissolved inorganic nitrogen (nitrite+nitrate) and phosphorus were observed in the area of the mauka spring water at the top of the farm. The average concentration for dissolved inorganic phosphorus (DIP) was 2.39 μmol at the mauka spring area compared with 1.54 μmol at the middle spring area and 1.53 μmol at the furthest point, the makai area (Fig 4A). Dissolved inorganic nitrogen (nitrite+nitrate, DIN) was 23.75 μmol at the mauka spring area, compared with 21.12 μmol at the middle spring, and the lowest concentration observed was at the makai spring area at 16.34 μmol (Fig 4B).

All standard curves for qPCR assay displayed high correlation coefficients ($R^2 = 0.99$) and similar slopes (S2 Fig). In addition, the efficiencies for amplification of the *16S*, *nirS*, *nifH*, and *amoA* genes was 95%, 99%, 96%, and 93%, respectively. In the watercress farm the average gene copies $mL^{-1}$ of the *16S* gene was $4.50x10^4$, $5.64x10^4$, and $1.04x10^6$ copies $mL^{-1}$ for the mauka, middle, and makai spring areas respectively. In comparison, the *16S* gene copies $ml^{-1}$ from water collected in the watercress plots averaged $1.81x10^6$ copies $ml^{-1}$, indicating there was greater biomass of Bacteria and Archaea in the watercress plots compared with the groundwater spring areas.

In this survey, the gene copy numbers (copies $ml^{-1}$) of the *nirS* gene were higher than those of *nifH* and *amoA*, which implies a greater abundance of N removal (nitrite reduction or denitrification) compared to N fixation or ammonia oxidation processes in the watercress farm. Abundances of nitrogen genes within the plots are averages of sample collections, which were sampled one to three times throughout the year. The largest abundances observed were denitfirication (*nirS*) genes in plots below the middle spring, B1 ($4.26x10^6$ copies $ml^{-1}$) and A2 ($4.06x10^6$ copies $ml^{-1}$; Fig 5) from plots with poor watercress health. Abundances of denitrification (*nirS*) genes in plots in the upper half of the farm associated with watercress in good health had $2.69x10^6$, and $2.65x10^6$ copies $ml^{-1}$ in S8 and R7, respectively (Fig 5). The largest abundances of *amoA* were measured in plot R7 ($2.08x10^6$ copies $ml^{-1}$) in good watercress condition, and B2 ($1.04x10^6$ copies $ml^{-1}$; Fig 5A qPCR) collected in a plot in poor condition. The gene *nifH* (nitrogen fixation) was observed at low abundances across all sites measured in the farm ranging from the lowest in the mauka spring ($1.65x10^2$ copies $ml^{-1}$) to the largest abundances in plot B1 ($1.01x10^5$ copies $ml^{-1}$) a plot in poor watercress health.

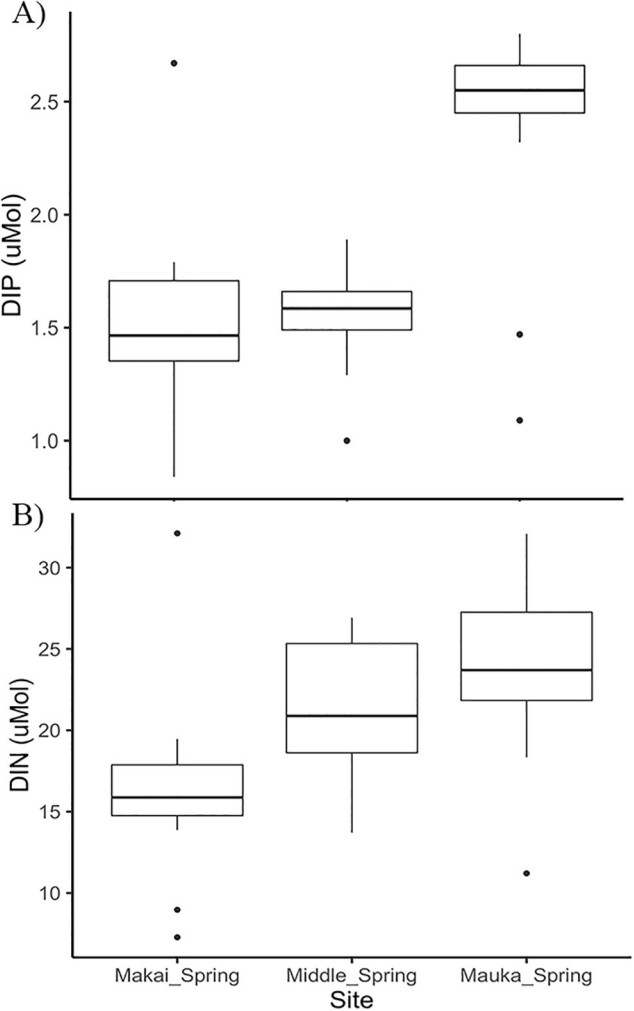

**Fig 4. Boxplots of A) dissolved inorganic phosphorus (μmol) and B) dissolved inorganic nitrogen (μmol) concentrations from surface water collected monthly for one year from the mauka, middle and makai springs in Sumida Farm.**

## Socio-cultural values of Kalauao Spring and Sumida Farm

Kalauao Spring, where Sumida Farm is located, has been highly used and valued for centuries. In our search of Papakilo, a Hawaiian language newspaper database, the place name search for "Kalauao" yielded nearly 500 hits [45]. A large majority of these refer to land claims and other legal documents, but some are moʻolelo, which are stories, histories, or accounts of places, people and deities passed down among generations [46]. These articles chronicle the importance of Kalauao Spring to people, agriculture, and culture, with accounts dating to as early as 1100 AD [47]. For example, one article attributed to Hawaiian historian Samuel Kamakau (1865) [20] speaks of the use of waters from Kalauao Spring for taro (kalo) spring-fed agriculture during the reign of Kalaʻimanuia in the 1100s:

> "...It was she [Kalaʻimanuia] who made Paʻaiau, Opu, and Kapaʻakea to be fishponds for herself; she also made large kalo terraces in Kalauao to supply herself with food. The land around Oʻahu yielded in abundance through much cultivation [during her time] ..."

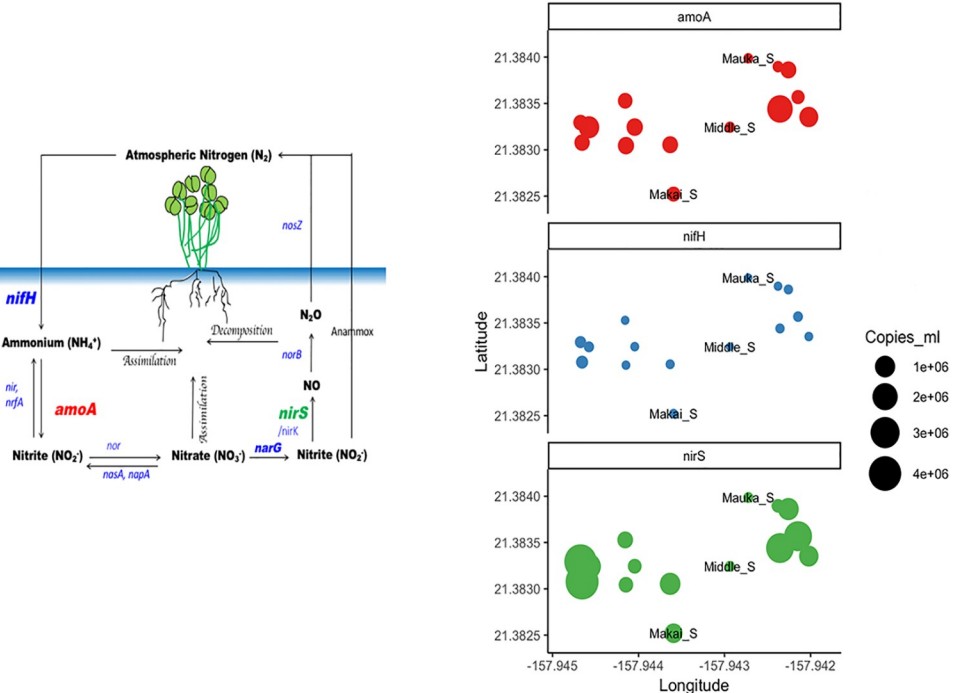

**Fig 5. Abundances (copies mL$^{-1}$) of the *amoA*, *nifH*, and *nirS* genes assayed from water samples collected from sites in the Sumida Farm.** Samples are grouped by gene and plotted on the Sumida Farm map (Fig 3) where samples were collected.

Similarly, in the story, "Nā 'Ano'ai Like 'Ole o 'Ewa," (1919) [22] the authors speak of the use of Kalauao Spring for taro production as well as swimming pools for chiefs:

> "I went to see the leaping place where chiefs would swim. It is very close to the water pump house of Kalauao. It was smooth and deep, and the name of this pool of water is Kahuawai. On the east side are some irrigated food terraces; it is a kind of pond that was somewhat deep in the old days. These [food terraces] were the kalo terraces that Kaho would always plant from the time he reached maturity."

Another newspaper article, "The Story of Ka'ehuikimanōopu'uloa, The Shark Child of Kapukapu and Hōlei," (Fig 6; [48]) also tells of famed swimming pools in Kalauao preferred for their fresh cold water.

> ". . ..Then the group finally reached the waters of Kahuawai in Kalauao, and there they swam for a long time. The visitors felt nothing but admiration for the beauty and the coolness of the swimming pools that belonged to this chiefess . . ."

While Kalauao Spring has changed substantially and now primarily supports watercress rather than taro, Sumida Farm, one of the oldest multi-generational farms remaining in the state, continues to have high social and cultural value [32]. The farm has been featured in a variety of community, local, and national newspapers and press [30–39], which highlight the value of the farm for local food production, multi-generational farming, and heritage and

**Fig 6. Excerpt of newspaper article from Ke Au Okoa, Vol. IV, Num. 34, Page 4, Columns 1–3 entitled "He Moʻolelo Kaao no Kaʻehuikimanōopuʻuloa," dated December 8, 1870.**

aesthetic values [33]. This includes being referred to as "a green patch of heaven" by the Honolulu Star Advertiser, the main newspaper for Hawaiʻi [34]. Community interest in watercress as a staple local crop, abundant in micronutrients, remains strong, particularly given recent growth of the farm-to-table local food movement [35]. For example, a feature in the Huffington Post details a famous local chef's impression of the farm [33]:

> "As we drove to Sumida Farms, Chef Mavro lit up as he told me about the farm which he described as the "most magical place on the island." Those are big words when you are talking about Hawaii, but if you are a chef who concentrates on local sourcing, then a 86-year-old watercress farm seems like a good choice to give the badge of "most magical"

The farm has also been featured in many local outlets, including books (e.g. "Saving the Family Farm" in Stories of Aloha: Homegrown Treasures of Hawaiʻi [36], videos [38, 39], and newspapers [34]. For example, the Hawaiʻi Independent, a local newspaper, featured the farm, concluding [37]:

> "The Sumida Farm also reminds us of how suburban land can be preserved for farming, but not without a fight. Finally, tours such as this one organized by Slow Foods show the potential of successful working farms like the Sumidas' are a model for how local agriculture, education, and agricultural tourism can converge to inspire future stewards of our land."

The farm also has value as a role model for other small farms in the community. They are known for good labor practices and sustainable integrated pest management practices, providing inspiration to the emerging diversified post-plantation farming sector in Hawaiʻi which aims to be more environmentally sustainable and socially just [37].

The heritage, social connection, and sense of place values of Sumida Farm are what motivate the Sumidas to continue farming despite the challenges they face. When Sumida Farm began in 1928, families of diverse ethnic backgrounds were farming a variety of wetland crops including ong choi, banana, and watercress. Barbara Sumida described how her grandparents began to farm watercress in the area:

> "Sumida grandma was the one, she was the boss. She was a real business lady and I think that's what inspired them to try something else [other than dairy farming]. There were some other families already growing watercress because they saw spring water and figured it was

*perfect for growing watercress. I don't know how they got the idea, where the watercress came from or how it got here . . .but Sumida obaa-chan [grandma] started to look for another place to farm and somehow found this place to rent."*

Barbara and David Sumida emphasized that by the time they were kids, the farm provided a solid livelihood for their family:

*". . .once my parents really got it going as a watercress farm, it did well enough to send us four kids to college."*

Today Sumida Farm is the largest of only five remaining watercress farms in the state (down from twelve in the 1960s), and crop yields are declining. However, the community, social, heritage, and educational values have grown. From the perspective of the Sumida family, the farm has great heritage value:

*"There's a big emotional attachment to this spot here cuz we grew up here, our parents worked here, and even our kids, even though they didn't really grow up here on the farm, they're starting to realize that they also have a strong tie to the farm."*

The farm is also highly valued by the broader community as one of the few remaining green spaces in Pearl Harbor:

*"In the 70s, you know all these condos, it was just built all really quickly, and so I think the community values, have this sense of ownership over this spot too. It's very important to the community, especially the old timers that remember when it was all sugarcane. . ."*

Over the past several years the Sumidas invited over 2000 visitors from schools to tour the farm and learn about where their food comes from, an experience few urban-dwelling Central Oʻahu students currently have:

*"We would always send them home with some watercress. Some of the teachers would take it another step and they'd have a little cooking class, sometimes they would go visit the supermarket, so the kids could see, "ok, this is where it grows, these are the men who work in the fields, they harvest it and make the bunch, then you see the bunch in the store, you take the bunch home and make your soup or whatever".*

While the Sumidas have invited people to the farm because they want to share their place, this has created strong community support, which they believe may help to ensure the renewal of their lease with the landowner:

*"Yeah, besides the value of the crop, it has the social value, it's probably more valuable than the crop . . .even that shack! [grass shack on farm property] If we were to tear down that shack over there, the outrage!"*

Yet, the Sumidas and other watercress farms have faced a number of challenges, the first being several key pest outbreaks. One major pest outbreak, the diamondback moth, nearly wiped out the watercress in the early 1980s, but a discovery led John McHugh to install a sprinkler system that saved the crop [26]. David Sumida explained how the diverse skill sets the family brought to the table ultimately led to the survival of the farm:

*"We all had skills and back then Barbara's husband, John McHugh, our entomologist, he became the general manager and came up with the idea of the sprinkler system. It worked really—it's just incredible. There was one thing our dad was struggling with—the diamondback moth. The diamondback moth was eating the whole field . . .by 1982 there was hardly any watercress left to harvest."*

More recently (in 2002, Fig 2) the aster yellows plant disease has also intermittently affected their crops, as has watercress rot.

Changes in water quality and quantity are also perceived to have affected the yields of the watercress. The Sumidas emphasized that the quality of the water is important.

*"It has to be clean spring water. You don't want to use stream water because it might . . .you don't know what's in that water, it's not clean. It could have parasites. . ."*

They perceived that declines in the groundwater resource had increased the salinity of spring discharge, leading to the majority of watercress farms going out of business.

*"Back in the day there was a lot more water and that's why there were more farms. As the water turned brackish because of the groundwater being used, that's why there's not as many farms now. Our dad told us at one time there was about 10 million gallons of fresh water coming through the land per day, then down to about 5 million gallons a day."*

Looking into the future, members of the Sumida family have different visions, but they all sense the need to adapt to changing conditions. David Sumida spoke of his desire to continue farming in the way they always had:

*"I'm really sentimental. I want the farm to stay just the way it is, like, for a long time. I don't really want it to change. . .. That's one of my favorite things to work alongside the workers. It's just, just, just . . .so much mana [Hawaiian word for spiritual energy of power and strength] out there. If you out there and you working, you forget about everything around you . . .the highway being here and then the rail. Forget about all these things. Powerful, lots of mana."*

Whereas, his sister Barbara emphasized that the next generation would make some changes and that the farm would continue to adapt:

*"We've all kind of been really lucky that we've managed to get through every disaster that came along, whether it was the diamondback moth or the aster yellows and now they're changing the food safety laws and, we've always been optimistic in that way, we've always thought, 'we'll figure it out—how to do it'. Now the next generation needs to figure out how to do it, because they have a different view that I don't have . . .and I think the community would be really mad if this got bulldozed over."*

## Discussion and conclusions

In a densely urbanized landscape, Sumida Farm reminds us of the springs that have flourished in the area for generations and shaped traditional and more recent agricultural practices. Hawaiian language newspaper translations [20, 22, 48] and interviews with the Sumida family demonstrate a landscape of continuous change around Kalauao Spring, from an area dominated by taro fields and natural pools where chiefs swam, to a mix of wetland crops, to the

current watercress farm. The Sumidas' adaptive strategies to counter previous and current challenges to crop production have changed as well. As wetland agricultural systems in urban settings become more rare, and interest in protecting and restoring them becomes higher, learning from the factors that have contributed to the Sumidas' persistence is critical and sheds light on the broader question of how urban ecosystems can thrive [49–51].

Analysis of 25 years of harvest, climate, and pest occurrence data emphasizes the potential importance of threshold effects in the success of spring-fed agricultural systems such as Sumida Farm, particularly in the face of expected climate-induced impacts to crops in the future [52, 53]. For example, though temperature effects on watercress production are now relatively small and confined to the hottest summer months, it is likely that climate change will eventually push monthly minimum and mean temperatures higher than the threshold values that affect crop production for more of the year, resulting in further declines in provisioning capability. Comparable threshold effects may be important for future precipitation, ONI conditions, and pest infestations as well. Among future management strategies, the Sumidas have discussed transitioning to heat and salt tolerant species of watercress, as has been done by a nearby watercress farm that is already experiencing saltwater intrusion into their water source.

Our correlation analysis indicates that the relationship between watercress production and groundwater pumping has been positive since 1994. However, because the underlying hydrogeological structure of the aquifer is still largely unknown [23], and because spring discharge measurements in the area are intermittent and widely distributed, it is difficult to directly determine the effect of regional pumping on regulating local spring discharge, and ultimately watercress production. Two discrete clusters of spring water on the farm have overlapping oxygen stable isotope values suggesting a similar recharge origin. Yet differences in salinity, nitrogen and phosphorus concentrations between springs separated by only 100–200 m suggest different aquifer flowpaths, or isolation by a confining layer of the groundwater paths feeding the two different groups of springs. This study did not reveal any seasonal or tidal changes in spring salinity, but the salinity of 1.2 in the middle springs suggests sensitivity to saltwater intrusion into the aquifer. Unregulated groundwater withdrawal, sea level rise, and/or decrease in recharge can induce upward and landward movement of saltwater resulting in springs becoming saltier over time [54]. As such, salinity in the middle spring, which is the highest of all the studied springs, should be subject to long-term monitoring to alert against seawater intrusion due to anthropogenic or climatic changes to aquifer conditions.

Surprisingly, data from intensive year-long sampling indicates high water quality in the spring water feeding Sumida Farm, despite the high degree of urban development in the surrounding area. The nutrient and stable isotope values of nitrate in the springs suggest only modest anthropogenic inputs, as they are comparable to levels in wells located upstream of developed areas on the island [55], on the eastern side of the Pearl Harbor aquifer, and also on the eastern side of Oʻahu [56]. These are in contrast to nutrient concentrations found in the predominantly agricultural western side of the Pearl Harbor aquifer, which has up to an order of magnitude higher nitrate and phosphate levels [57]. All measured wastewater tracers were below detection limits suggesting no direct effluent leakage from cesspools and sewer lines into groundwater feeding the springs. However, it is expected that due to ongoing sea level rise, at 1 m higher sea level, sewer mains and on-site sewage disposal systems in the vicinity of the Sumida Farm will be chronically flooded [58], which may result in wastewater leakage to the surrounding aquifer, and contamination of the springs.

Overall high abundances of denitrifiers at Sumida Farm suggest that the watercress farm is providing water quality protection by removing bioreactive N [59, 60], thereby providing an

important nutrient retention service, such as those provided by natural and constructed wetlands [61]. Understanding N cyclers in the watercress farm may not directly answer questions regarding decades-long declining yield, however, microbes act as sentinels of unhealthy water quality or ecosystem health [62], which may explain our observation of high abundances of denitrifiers in poor health plots in the lower half of the farm [63]. Denitrification is a natural process occurring in watercress farms, yet removal of bioavailable N by microbes puts them in direct competition with plant uptake and growth. One strategy to regulate denitrification buffer zones might include increased cleaning of beds to keep organic matter from accumulating and allowing for maximum water flow and aeration (oxygen) through the plots. Further, the dead, decaying watercress plant material may act as a good fertilizer (due to its high N content) if mulched and spread thinly throughout plots in early growth stages.

The Sumidas' relationships with the broader community have enhanced their ability to adapt and innovate in response to challenges and changing conditions. Today, the Sumidas' role as partners in the current study allows them to draw on data from different fields to better understand how their farm can continue to thrive going forward. Beyond measures that directly support crop production into the future, the next generation of Sumidas are considering more substantive changes to farm operations, including a transition to primarily demonstration farming and farm tours, with a focus on education about traditional agriculture and the cultural heritage of the Kalauao Spring site [14]. This aligns with the vision of the landowner, who has actively solicited alternate business models from the Sumida family as part of lease renegotiations. There is clear will for this system to thrive based on multiple motivations, including economic and local food production benefits, community and social values (e.g. nostalgia for an agricultural past), and recognition of the nitrogen retention services of systems such as their farm. Community recognition of the numerous non-monetary benefits provided by the farm creates opportunities to support evolving farm operations into the future.

## Supporting information

**S1 Table. Primer sets used for quantitative PCR.**
(PDF)

**S2 Table. Chemical parameters measured in selected springs at Sumida Farm.** Abbreviations in the table include: Radon, Rn; Salinity, Sal; Specific Conductivity, SPC; Dissolved Oxygen, DO; Temperature, Temp; Caffeine, Caf; ethynylestradiol, EE2; Atrazine, Atr; and Dicholor-diphenyl-trichloroethane and its degradation product DDE, DDT. Stable isotopes of oxygen, δ18O; hydrogen, δD; Standard deviation of both isotopes, SD δ18O and SD δD; Total dissolved nitrogen, Total N; Total dissolved phosphorus, Total P; dissolved inorganic P, P; Nitrite+Nitrate, N+N; Isotopes of 15N- as NO3-, δ15N-NO3-, and oxygen as -NO3-, δ18O-NO3-. Detection limits, dl include: Caffeine dl = 150 ng/L; Carbamazepine dl = 25 ng/L; EE2 dl = 50 ng/L; Atrazine dl = 50 ng/L;, DDT dl = 0.6 ng/L.
(PDF)

**S1 Fig. Sumida Farm and surrounding groundwater pumping wells in the Waimalu groundwater management unit (black dot indicates location of Sumida Farm; red dots denote pumping wells strongly weighted in the inverse-distance weighted sum; purple dots denote pumping wells weakly weighted in the inverse-distance weighted sum).**
(PDF)

**S2 Fig. Standard curves of 16S, amoA, nifH, and nirS qPCR assays acquired by plotting gene copy number (log copies) by threshold cycle (Ct).** Equations for 16S: y = -3.4621x + 38.896;

amoA: y = -3.5148x + 47.645; nifH: y = -3.4341x + 36.174; and nirS: y = -3.3436x + 43.568.
(PDF)

**S3 Fig.**
(PNG)

## Acknowledgments

This manuscript is dedicated to the wonderful Barbara Sumida, the inspiration of this work, whom we all greatly miss. Barbara Sumida passed away before the submission of the final version of this manuscript. Dr. Jennifer Engels accepts responsibility for the integrity and validity of the data collected and analyzed. This project would not have been possible without the generosity, knowledge, and welcome of the extended Sumida family: Barbara, David, and Stephen Sumida; John McHugh, and Emi and Kyle Suzuki. We gratefully acknowledge the Hawaiian language translation work of John Jacob Chock and Jeffrey "Kapali" Lyon of the Institute of Hawaiian Language Research and Translation, and the interview provided by the Groundwater division of the Commission on Water Resource Management. We also thank Gregory Chun and Piʻilani Smith for their involvement in the project, and Piʻilani Smith for his work transcribing interviews.

## Author Contributions

**Conceptualization:** Jennifer L. Engels, Sheree Watson, Henrietta Dulai, Kimberly M. Burnett, ʻAnoʻilani Aga, John McHugh, Barbara Sumida, Leah L. Bremer.

**Formal analysis:** Jennifer L. Engels, Sheree Watson, Henrietta Dulai, ʻAnoʻilani Aga, Nathan DeMaagd.

**Funding acquisition:** Jennifer L. Engels.

**Investigation:** Jennifer L. Engels, Sheree Watson, Henrietta Dulai, Kimberly M. Burnett, ʻAnoʻilani Aga, John McHugh, Barbara Sumida, Leah L. Bremer.

**Methodology:** Jennifer L. Engels, Sheree Watson, Henrietta Dulai, Kimberly M. Burnett, Christopher A. Wada, ʻAnoʻilani Aga, John McHugh, Barbara Sumida, Leah L. Bremer.

**Project administration:** Jennifer L. Engels, ʻAnoʻilani Aga.

**Supervision:** Jennifer L. Engels, Kimberly M. Burnett, Christopher A. Wada, ʻAnoʻilani Aga.

**Validation:** Henrietta Dulai, ʻAnoʻilani Aga, Nathan DeMaagd, John McHugh, Barbara Sumida, Leah L. Bremer.

**Visualization:** Sheree Watson, Kimberly M. Burnett, Christopher A. Wada, Nathan DeMaagd.

**Writing – original draft:** Jennifer L. Engels, Sheree Watson, Henrietta Dulai, Kimberly M. Burnett, Christopher A. Wada, Leah L. Bremer.

**Writing – review & editing:** Sheree Watson, Kimberly M. Burnett, Christopher A. Wada, ʻAnoʻilani Aga, Nathan DeMaagd, John McHugh, Barbara Sumida, Leah L. Bremer.

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
