## [Decision Letter · Decision Letter 0]

30 Sep 2019

PONE-D-19-21079

Collaborative research to assess resilience of urban green infrastructure: the case of the Sumida watercress farm near Pearl Harbor, Oʻahu

PLOS ONE

Dear Dr. Jennifer,

Thank you for submitting your manuscript to PLOS ONE. After careful consideration, we feel that it has merit but does not fully meet PLOS ONE’s publication criteria as it currently stands. Therefore, we invite you to submit a revised version of the manuscript that addresses the points raised during the review process.

We would appreciate receiving your revised manuscript by October 30, 2019. To enhance the reproducibility of your results, we recommend that if applicable you deposit your laboratory protocols in protocols.io, where a protocol can be assigned its own identifier (DOI) such that it can be cited independently in the future. For instructions see: http://journals.plos.org/plosone/s/submission-guidelines#loc-laboratory-protocols

We look forward to receiving your revised manuscript.

Kind regards,

Weili Duan, Ph.D

Academic Editor

PLOS ONE

**Journal Requirements:**

3. Please upload a new copy of Figure 4 and 5 as the detail is not clear. Please follow the link for more information:  http://blogs.plos.org/everyone/2011/05/10/how-to-check-your-manuscript-image-quality-in-editorial-manager/ and and this link for our full figure guidelines http://www.plosone.org/static/figureGuidelines.

4.We note that you have included the phrase “data not shown” in your manuscript. Unfortunately, this does not meet our data sharing requirements. PLOS does not permit references to inaccessible data. We require that authors provide all relevant data within the paper, Supporting Information files, or in an acceptable, public repository. Please add a citation to support this phrase or upload the data that corresponds with these findings to a stable repository (such as Figshare or Dryad) and provide and URLs, DOIs, or accession numbers that may be used to access these data. Or, if the data are not a core part of the research being presented in your study, we ask that you remove the phrase that refers to these data.

**Comments to the Author**

1. Is the manuscript technically sound, and do the data support the conclusions?

Reviewer #1: No

Reviewer #2: No

2. Has the statistical analysis been performed appropriately and rigorously? 

Reviewer #1: I Don't Know

Reviewer #2: Yes

3. Have the authors made all data underlying the findings in their manuscript fully available?

Reviewer #1: No

Reviewer #2: No

4. Is the manuscript presented in an intelligible fashion and written in standard English?

Reviewer #1: No

Reviewer #2: Yes

5. Review Comments to the Author

Reviewer #1: The paper is intended to describe an example of resilient urban social-ecological system that provides a suite of ecosystem services and other benefits, i.e. Sumida farm in Oahu, Hawaii. The farm survived during the years thanks to multiple factors, including economic and local food production benefits, community and social values. The recognition of the nitrogen retention and possibly flood prevention services of systems contributed to its resilience. However, in the face of changing climate, ongoing urbanization, agricultural pests and diseases, and increasing demands on the Pearl Harbor aquifer, Sumida Farm is likely to bear escalating stressors in the decades to come. Data derived from the farm allowed to better understand how crop yields can be sustained going forward.

The paper is interesting and offers a good example of collaboration between researches and a local community (the Sumida family). However, major weaknesses prevent from its publication:

- In my conception, a farm is not “green infrastructure”; if so, authors should define their notion of “green infrastructure” at the beginning of the paper, with appropriate references.

- The structure of the paper is confused and the paper too long. The aim of the paper is unclear: is this identifying the resilience factors of the farm? Or the quality and quantity of the spring water? This should be made clear and remain consistent in the whole manuscript. Conclusion should “match” the identified aim.

- “resilience” is mentioned many times in the text. However, it is not explained what resilience means in this paper (e.g. resilience to what?), as well as the past/present/future “challenges” faced by the farm.

Further comments are listed below:

L47/56: what do authors mean for resilience? It would be interesting to know why other farms did not survive over the year (with some numbers and evidence in support).

L123-124: objective 1 represents more a methodological step, rather than a research objective

L125-126: what is this analysis for?

L130: could be a farm considered a piece of “green infrastructure”?

L142(and successive): join with L82-91 (and cut)

L185(and successive): the section text does not reflect the title of the section.

L208: what is the family interest in this piece of research?

L320-475: this section does not belong to results, since this is still about the history of the farm (and various other stories); this information should be summarised in a more succinct way.

L446: how did the Sumida family adapt?

L590: how will climate change affect the farm, on the basis of projections and/or NOAA information? How is the family preparing for it, considering the uncertainties of such projections/information? Are other farms taking example from the Sumida family? How could this model be exported in other farms/islands? Which was the main factor(s) of resilience of the Sumida farm, based on obtained results?

My recommendation is: re-submit, major revision.

Reviewer #2: The key issue for me with the paper is that there is no underpinning theorisation of "resilience" against which the mixed methods results can be assessed. As a result I had a hard time making sense of the documentary and interview sources on the one hand and the quantitative sources on the other. Both seem to be important pieces of a larger puzzle -- a quant-qual approach to theorising "resilience"??-- but are not clearly framed as such here. In the absence of this critical foundation, assertions about the resilience of Sumida Farm seem only superficially true (i.e. it is resilient because it has been around for a while).

There does seem to be a story about resilience to tell here though, and the authors bring to the table all the ingredients for a compelling case study -- which is why I do not recommend rejection of the submission. Rather, I think that a revised version of the paper that included a new section of perhaps 1000 words proposing a quant-qual conceptualisation of resilience that matches the quant-qual data set would be much stronger and would indeed greatly interest the PLOS One readership. In this context I note that you include Wolch et al (2014) in your reference list -- why not take up their invitation to see greened/greening landscapes as part of racialised urban spaces? Not only does this seem a potentially useful avenue from the point of view of capitalist urban political economy (the story of the transformation of the Pearl Harbour wetland from natural wetland/agroecological complex to urban sprawl), but it also explicitly recognises the manipulation of non-eurocentric geographies as part of this accumulation logic. Possibly this would entail some more work, particularly in terms of interviews with contemporary native Hawaiian activists (whose voices are notable absent from the current version), but this could be done quickly. For example, it is hinted that perhaps there may be a problem with renewal of the lease for Sumida Farm.....is this because there are alternative perspectives on the best use for the site? And if so from whom? What do native Hawaiian representatives say? Capturing some of this information might also allow for clearer linkages with the historic-documentary material presented earlier and by way of background to the Pearl Harbour wetland.

6. PLOS authors have the option to publish the peer review history of their article (what does this mean?). If published, this will include your full peer review and any attached files.

Reviewer #1: No

Reviewer #2: No

---

## [Author Response · Author response to Decision Letter 0]

22 Jan 2020

Jennifer Engels

Hawai‘i Institute of Geophysics and Planetology

University of Hawai‘i at Mānoa

1680 East-West Road

Honolulu, HI 96822

engels@hawaii.edu

January 19, 2020

Dear Dr. Duan:

We thank you and the two reviewers for your constructive and helpful feedback to our manuscript, now entitled “Collaborative research to support urban agriculture in the face of change: the case of the Sumida watercress farm near Pearl Harbor, Oʻahu” by Jennifer Engels, Sheree Watson, Henrietta Dulai, Kimberly Burnett, Christopher Wada, ‘Ano‘ilani Aga, Nathan DeMaagd, John McHugh, Barbara Sumida, and Leah Bremer.. We have carefully considered each recommendation and have substantially improved the paper. The revised manuscript is now a compelling case study of the supports and challenges for urban agriculture in a changing environment. 

Below are the [editor and reviewer comments] and our responses.

[Thank you for submitting your manuscript to PLOS ONE. After careful consideration, we feel that it has merit but does not fully meet PLOS ONE’s publication criteria as it currently stands. Therefore, we invite you to submit a revised version of the manuscript that addresses the points raised during the review process.

We would appreciate receiving your revised manuscript by October 30, 2019. ]

On October 21, 2019, we requested (and were granted by PLOS ONE editorial staff) an extension to the re-submission timeline for the following reasons:

1) Our 10 co-authors include two farmers, the head of a state agency, and researchers located across 3 time zones. We worked intently and continuously on the revisions since receipt of the Editor's request for revisions, but due to our varied schedules and locations it was difficult for us to finish everything by the original re-submission deadline of October 30, 2019.

2) The extent of the revisions requested by both reviewers was "major". In addition, while some of the reviewers' comments overlapped, many of their suggestions for revision were widely divergent, necessitating a series of decisions on our part about if and how best to reconcile their requests.

We are grateful for the opportunity to re-submit, and think that we are now returning the best possible manuscript that addresses all reviewers' concerns.

[]

Please see our updated financial disclosure statement in the last paragraph of the new cover letter.

[To enhance the reproducibility of your results, we recommend that if applicable you deposit your laboratory protocols in protocols.io, where a protocol can be assigned its own identifier (DOI) such that it can be cited independently in the future. For instructions see: http://journals.plos.org/plosone/s/submission-guidelines#loc-laboratory-protocols]

Our laboratory protocols are not novel. They are all previously published, and we referenced the kits and the companies where the kits were purchased which include standard protocols/instructions in the Methods descriptions. Supplementary information Tables S1, and Tables S2 provide data needed to replicate qPCR methods, standards data to assure efficiencies reported in the manuscript, and the geochemistry data as reported in the manuscript. 

[Please include the following items when submitting your revised manuscript:

• A rebuttal letter that responds to each point raised by the academic editor and reviewer(s). This letter should be uploaded as separate file and labeled 'Response to Reviewers'.

• A marked-up copy of your manuscript that highlights changes made to the original version. This file should be uploaded as separate file and labeled 'Revised Manuscript with Track Changes'.

• An unmarked version of your revised paper without tracked changes. This file should be uploaded as separate file and labeled 'Manuscript'.]

Done.

[Please note while forming your response, if your article is accepted, you may have the opportunity to make the peer review history publicly available. The record will include editor decision letters (with reviews) and your responses to reviewer comments. If eligible, we will contact you to opt in or out.

We look forward to receiving your revised manuscript.

Kind regards,

Weili Duan, Ph.D

Academic Editor

PLOS ONE

Journal Requirements:

http://www.journals.plos.org/plosone/s/file?id=wjVg/PLOSOne_formatting_sample_main_body.pdf and http://www.journals.plos.org/plosone/s/file?id=ba62/PLOSOne_formatting_sample_title_authors_affiliations.pdf]

All files are correctly formatted and all files are correctly named.

[2. We note that you have stated that you will provide repository information for your data at acceptance. Should your manuscript be accepted for publication, we will hold it until you provide the relevant accession numbers or DOIs necessary to access your data. If you wish to make changes to your Data Availability statement, please describe these changes in your cover letter and we will update your Data Availability statement to reflect the information you provide.]

All relevant data are included either in the Manuscript, the references, or in the Supplementary information. As such there is no need for separate repository information.

[3. Please upload a new copy of Figure 4 and 5 as the detail is not clear. Please follow the link for more information: http://blogs.plos.org/everyone/2011/05/10/how-to-check-your-manuscript-image-quality-in-editorial-manager/ and and this link for our full figure guidelines http://www.plosone.org/static/figureGuidelines.]

Prior Figures 4 (new Figure 3) and 5 (new Figure 4) have been updated using the PACE tool referenced from the web link listed above. 

[4. We note that you have included the phrase “data not shown” (line 573) in your manuscript. Unfortunately, this does not meet our data sharing requirements. PLOS does not permit references to inaccessible data. We require that authors provide all relevant data within the paper, Supporting Information files, or in an acceptable, public repository. Please add a citation to support this phrase or upload the data that corresponds with these findings to a stable repository (such as Figshare or Dryad) and provide and URLs, DOIs, or accession numbers that may be used to access these data. Or, if the data are not a core part of the research being presented in your study, we ask that you remove the phrase that refers to these data.]

All data are now either in the Manuscript, the references, or the Supplementary information, and the phrase “data not shown” in prior line 573 has been removed.

[5. Review Comments to the Author

Reviewer #1: The paper is intended to describe an example of resilient urban social-ecological system that provides a suite of ecosystem services and other benefits, i.e. Sumida farm in Oahu, Hawai‘i. The farm survived during the years thanks to multiple factors, including economic and local food production benefits, community and social values. The recognition of the nitrogen retention and possibly flood prevention services of systems contributed to its resilience. However, in the face of changing climate, ongoing urbanization, agricultural pests and diseases, and increasing demands on the Pearl Harbor aquifer, Sumida Farm is likely to bear escalating stressors in the decades to come. Data derived from the farm allowed to better understand how crop yields can be sustained going forward.

The paper is interesting and offers a good example of collaboration between researchers and a local community (the Sumida family). However, major weaknesses prevent from its publication:]

Response R1.1. Thank you for your constructive and helpful comments. We have made substantial changes that address each of the comments below. 

[- In my conception, a farm is not “green infrastructure”; if so, authors should define their notion of “green infrastructure” at the beginning of the paper, with appropriate references.]

Response R1.2: This is a good point, and we have now shifted the focus from green infrastructure to urban ecosystems, including agricultural systems. Prior references to green infrastructure in the Abstract (lines 37 and 39), Introduction (lines 62, 67, 69, 72, 78, 83, 130), and Discussion and conclusions section (line 598) have all been replaced. While there are examples of farms being considered a type of green infrastructure in the literature (particularly those with specific best management practices), we agree with the reviewer that this detracts from the main messages of the paper and that urban ecosystems is a better term. 

[- The structure of the paper is confused and the paper too long. The aim of the paper is unclear: is this identifying the resilience factors of the farm? Or the quality and quantity of the spring water? This should be made clear and remain consistent in the whole manuscript. Conclusion should “match” the identified aim.]

Response R1.3: Thank you for pointing out that the aim of our article was unclear. We have reduced the length of the overall manuscript, and re-organized and restructured the objectives and entire manuscript it to be more clear. The paper is now organized around the following research questions which are first presented in the Introduction:

1) What environmental factors may have influenced watercress yields over the past 25 years?

2) What is the current quality and quantity of spring water on the farm and how does it relate to the farm's ability to continue to produce watercress?

3) From the perspective of the farming family, and current and historical narratives, what additional benefits are associated with Sumida Farm and Kalauao Spring that may also influence the farm’s persistence over time?

Each subsequent section of the paper (Methods, Results, Discussion and conclusions) is now organized around the three research questions.

[- “resilience” is mentioned many times in the text. However, it is not explained what resilience means in this paper (e.g. resilience to what?), as well as the past/present/future “challenges” faced by the farm.]

Response R1.4: We agree that the word “resilience” was not well defined in the initial version of our manuscript. We decided instead to simplify and shift the focus of the paper to examining the ways in which the farm persisted and adapted in the face of previous challenges to watercress production in the decades leading up to our analyses. We then analyze and discuss the factors that we think will continue to be key to the farm’s future sustainability based on our research, from the perspective of climate change, urbanization, and community values. The research is organized around our three questions (see Response R1.3 above), rather than around one concept like “resilience.” 

[Further comments are listed below:

L47/56: What do authors mean for resilience? It would be interesting to know why other farms did not survive over the years (with some numbers and evidence in support).]

Response R1.5: We are no longer using the phrasing “resilience” (see response R1.4 above). 

Based on interviews with the Sumida family, we now include information in the Results (line 474-475) saying that “today Sumida Farm is the largest of only five remaining watercress farms in the state (down from twelve in the 1960s).” We also include a second quote from David Sumida saying “Back in the day there was a lot more water and that’s why there were more farms. As the water turned brackish because of the groundwater being used, that’s why there’s not as many farms now. Our dad told us at one time there was about 10 million gallons of fresh water coming through the land per day, then down to about 5 million gallons a day.” (lines 532-536). Finally, we reference an adaptation strategy already in use by one of the other remaining watercress farms in the Discussion and conclusions section (lines 595-597) saying “the Sumidas have discussed transitioning to heat and salt tolerant species of watercress, as has been done by a nearby watercress farm that is already experiencing saltwater intrusion into their water source.” 

[L123-124: objective 1 represents more a methodological step, rather than a research objective]

Response R1.6: We removed this objective and the other two objectives from the original manuscript. The paper is now organized around the following research questions which are first presented in the Introduction and referred to throughout the manuscript:

1) What environmental factors may have influenced watercress yields over the past 25 years?

2) What is the current quality and quantity of spring water on the farm and how does it relate to the farm's ability to continue to produce watercress?

3) From the perspective of the farming family, and current and historical narratives, what additional benefits are associated with Sumida Farm and Kalauao Spring that may also influence the farm’s persistence over time?

Each subsequent section of the paper, (Results, Discussion and conclusions) is now organized around the three research questions.

[L125-126: what is this analysis for?]

Response R1.7: We removed this objective and the other two objectives from the original manuscript. The paper is now organized around the following research questions which are first presented in the Introduction:

1) What environmental factors may have influenced watercress yields over the past 25 years?

2) What is the current quality and quantity of spring water on the farm and how does it relate to the farm's ability to continue to produce watercress?

3) From the perspective of the farming family, and current and historical narratives, what additional benefits are associated with Sumida Farm and Kalauao Spring that may also influence the farm’s persistence over time?

Each subsequent section of the paper (Methods, Results, Discussion and conclusions) is now organized around the 3 research questions.

[L130: could a farm be considered a piece of “green infrastructure”?]

Response R1.8: We have shifted the focus from green infrastructure to urban ecosystems, including agricultural systems. Prior references to green infrastructure in the Abstract (lines 37 and 39), Introduction (lines 62, 67, 69, 72, 78, 83, 130), and Discussion and conclusions section (line 598) have all been replaced.

[L142 and successive: join with L82-91 and cut] 

Response R1:9 The introductory paragraph of the “Study site” portion of the Methods section of the paper (now lines 111-120) provides an important rationale for why the Sumida Farm/Kalauao Spring study site was chosen. Specifically, the long history of traditional and modern agriculture in the area and the unique cultural importance of the area were two deciding factors in our methodology for choosing the study site, knowing that we could collect both quantitative and qualitative information as result. As such we have kept this section intact, and hope that the readers will recognize the value of this information to the Methods section.

In the revised overall Introduction to the manuscript, we include more general contextual background as to why the Sumida Farm/Kalauao Spring area is a model system for answering questions about the survival of urban agricultural systems generally (lines 83-99). We also introduce here our co-production of knowledge approach, discuss the different ways that urban ecosystems are valued, and present the research questions of the paper. 

[L185 and successive: the section text does not reflect the title of the section]

Response R1.10: This section has been moved (starts at line 281) and substantially revised to make the methodology more clear. It is now titled “Social values of Sumida Farm that contribute to farm persistence - methods”. We describe our analysis of Hawaiian language newspaper archives, and how we conducted our semi-structured interviews with the State water regulator and the Sumida family.

[L208: what is the family interest in this piece of research?] 

Response R1.11: We clarified this portion of the Methods section (now starting on line 167) to describe how we worked together with the Sumida family farmers to compile their harvest records and annotate them using their knowledge of the reasons for crop failures in the past. This information was then utilized to determine which climate and other datasets (e.g. groundwater pumping) we should compare to the 25 years of crop harvest data in our correlation analysis to determine potential reasons for overall crop declines in the harvest record.

[L320-475: this section does not belong to results, since this is still about the history of the farm (and various other stories); this information should be summarised in a more succinct way] 

Response R1.12: We have modified the Introduction and Methods sections of the paper to make clear that collection of both quantitative (e.g. groundwater salinity measurements and microbial counts) and qualitative (e.g. cultural histories from Hawaiian language newspapers and interviews with the farmers) datasets are important components of our mixed methods approach to valuing the urban agriculture of Sumida Farm/Kalauao Spring. In particular, we created a new table (Table 1 “Assessment methodology - quantitative and qualitative factors”, line 181) that details the importance of each of these datasets to watercress yields on the farm. These changes to the manuscript should make it more clear to the reader that the Hawaiian language newspaper archive information and the interviews with the Sumida family and State water regulator are in fact important Results that help to explain the longevity of the watercress farm, as well as its prospects for surviving into the future. 

[L446: how did the Sumida family adapt?]

Response R1.13 We give various examples of the farmers’ previous adaptation strategies in the Results section. In one quote from Barbara Sumida (starting with line 552) she says “We’ve all kind of been really lucky that we’ve managed to get through every disaster that came along, whether it was the diamondback moth or the aster yellows and now they're changing the food safety laws and, we’ve always been optimistic in that way, we’ve always thought, ʻwe’ll figure it out--how to do it’.” In the “Discussion and conclusions” section, we give examples of planned future adaptation strategies: (line 595-597) “Among future management strategies, the Sumidas have discussed transitioning to heat and salt tolerant species of watercress, as has been done by a nearby watercress farm that is already experiencing saltwater intrusion into their water source.” and (line 648-651) “...the next generation of Sumidas are considering more substantive changes to farm operations, including a transition to primarily demonstration farming and farm tours, with a focus on educating Hawai‘i citizens about traditional agriculture and the cultural heritage of the Kalauao Spring site...” 

[L590: how will climate change affect the farm, on the basis of projections and/or NOAA information? How is the family preparing for it, considering the uncertainties of such projections/information?] 

Response R1.14: We clarified this section (now titled Discussion and conclusions) by explaining that temperature and other climate thresholds are likely to become more damaging with future climate change. For example, we say (starting on line 590) “though temperature effects on watercress production are now relatively small and confined to the hottest summer months, it is likely that climate change will eventually push monthly minimum and mean temperatures higher than the threshold values that affect crop production for more of the year, resulting in further declining yields.”

The family is preparing in various ways for future climate challenges, for example by considering a change to more salt and heat tolerant species of watercress (line 595-597), and modifying their watercress plot cleaning and mulching protocols (lines 637-642). In a more substantial shift, the next generation of farmers is considering a transition to primarily demonstration farming and farm tours (line 648-651). 

[Are other farms taking example from the Sumida family? How could this model be exported in other farms/islands?] 

Response R1.15: The Sumida family seem to be survivors from another era. As detailed in an article we cite (Suryanata K, Mostafanezhad M. Is farming sexy? Agro-food initiatives and the contested value of agriculture in post-plantation Hawai‘i. Geoforum. 2018;91.), most small farms in Hawai‘i today are newly-founded “agro-food initiatives” (farm tours, volunteer farming, farming supported by a working spouse), rather than actual for-profit farms. The Sumidas’ ability to weather a variety of challenges to crop production over the years and remain profitable makes them somewhat unique in Hawai‘i today. However, the Hawai‘i public’s interest in farming is on the rise (Suryanata and Mostafanezhad, 2018), and the Sumidas provide a highly visible and valued example of for-profit farming at a small scale, as demonstrated by Barbara Sumida’s quote in the Results section (line 556-557): “I think the community would be really mad if this got bulldozed over.” 

[Which was the main factor(s) of resilience of the Sumida farm, based on obtained results?]

Response R1.15: As detailed in R1.4 (above), we have removed the “resilience” framing of the paper. But in the spirit of the reviewer’s question, Barbara Sumida says in the Results section (line 552-555) “We’ve all kind of been really lucky that we’ve managed to get through every disaster that came along, whether it was the diamondback moth or the aster yellows and now they're changing the food safety laws and, we’ve always been optimistic in that way, we’ve always thought, ʻwe’ll figure it out--how to do it’.” The Sumidas’ adaptive strategies have served them well through the 90+ years of the farm’s existence, more than any single climate factor, pest outbreak, or change in land or resource use. And as we say in the Discussion and conclusions section (line 644-645): “The Sumidas’ relationships with the broader community have enhanced their ability to adapt and innovate in response to adverse events…” The Sumidas’ long-standing mutually-beneficial interaction with their community has had positive effects on their ability to adapt.

[My recommendation is: re-submit, major revision.

Reviewer #2: The key issue for me with the paper is that there is no underpinning theorisation of "resilience" against which the mixed methods results can be assessed. As a result I had a hard time making sense of the documentary and interview sources on the one hand and the quantitative sources on the other. Both seem to be important pieces of a larger puzzle -- a quant-qual approach to theorising "resilience"??-- but are not clearly framed as such here. In the absence of this critical foundation, assertions about the resilience of Sumida Farm seem only superficially true (i.e. it is resilient because it has been around for a while).

There does seem to be a story about resilience to tell here though, and the authors bring to the table all the ingredients for a compelling case study -- which is why I do not recommend rejection of the submission. Rather, I think that a revised version of the paper that included a new section of perhaps 1000 words proposing a quant-qual conceptualisation of resilience that matches the quant-qual data set would be much stronger and would indeed greatly interest the PLOS One readership.] 

Response R2.1: The term “resilience” has been extensively discussed and repeated defined over many decades in the literature (e.g. Folke, C. 2016. Resilience (Republished). Ecology and Society 21(4):44. https://doi.org/10.5751/ES-09088-210444). As such, we felt that it was beyond the scope of this paper to attempt a new conceptualization of resilience.

We have instead changed the focus of the paper to using quantitative and qualitative methods for examining the ways in which the farm survived previous challenges to watercress production in the decades leading up to our analyses. We then analyze and discuss the factors that we think will be key to the farm’s future sustainability, from the perspective of climate change, urbanization, and community values. The research is organized around our three questions, rather than around one concept like “resilience.” 

We have modified the Introduction and Methods sections of the paper to make clear that collection of both quantitative (e.g. groundwater salinity measurements and microbial counts) and qualitative (e.g. cultural histories from Hawaiian language newspapers and interviews with the farmers) datasets are important components of our mixed methods approach to valuing the urban agriculture of Sumida Farm/Kalauao Spring. We created a new table (Table 1 “Assessment methodology - quantitative and qualitative factors”, line 181) that details the importance of each of these datasets to watercress yields on the farm. These changes to the manuscript should make it more clear to the reader that the Hawaiian language newspaper archive information and the interviews with the Sumida family and State water regulator are in fact important Results that help to explain the longevity of the watercress farm as well as its prospects for surviving into the future.

[In this context I note that you include Wolch et al (2014) in your reference list -- why not take up their invitation to see greened/greening landscapes as part of racialised urban spaces? Not only does this seem a potentially useful avenue from the point of view of capitalist urban political economy (the story of the transformation of the Pearl Harbour wetland from natural wetland/agroecological complex to urban sprawl), but it also explicitly recognises the manipulation of non-eurocentric geographies as part of this accumulation logic. Possibly this would entail some more work, particularly in terms of interviews with contemporary native Hawaiian activists (whose voices are notably absent from the current version), but this could be done quickly. For example, it is hinted that perhaps there may be a problem with renewal of the lease for Sumida Farm.....is this because there are alternative perspectives on the best use for the site? And if so from whom? What do native Hawaiian representatives say? Capturing some of this information might also allow for clearer linkages with the historic-documentary material presented earlier and by way of background to the Pearl Harbour wetland.]

Response R2.2: We think that the reviewer’s point here is a fascinating and worthwhile direction for future research, but beyond the scope of the current project. We do clarify in the new version of the Discussion and conclusions that the next generation of Sumida farmers have plans for more substantive changes to farm practices, including a shift to primarily demonstration farming and farm tours, which “aligns with the vision of the landowner, who has actively solicited alternate business models from the Sumida family as part of lease renegotiations.” (line 651-653) 

The extensive rewrite we have undertaken of the original manuscript has streamlined, focused, and clarified the objectives and outcomes of our study of the Sumida Farm/Kalauao Spring area. At this point, we believe it will make an excellent contribution to the special issue Urban Ecosystems in PLOS ONE.

Thank you very much for your consideration.

Sincerely,

Jennifer Engels

Affiliate Faculty, Hawai‘i Institute of Geophysics and Planetology

---

## [Decision Letter · Decision Letter 1]

5 Mar 2020

PONE-D-19-21079R1

Collaborative research to support urban agriculture in the face of change: the case of the Sumida watercress farm near Pearl Harbor, Oʻahu

PLOS ONE

Dear Dr. Jennifer,

Thank you for submitting your manuscript to PLOS ONE. After careful consideration, we feel that it has merit but does not fully meet PLOS ONE’s publication criteria as it currently stands. Therefore, we invite you to submit a revised version of the manuscript that addresses the points raised during the review process. 

Please carefully deal with the comments from the second reviewer.

We would appreciate receiving your revised manuscript by April 3, 2020. To enhance the reproducibility of your results, we recommend that if applicable you deposit your laboratory protocols in protocols.io, where a protocol can be assigned its own identifier (DOI) such that it can be cited independently in the future. For instructions see: http://journals.plos.org/plosone/s/submission-guidelines#loc-laboratory-protocols

We look forward to receiving your revised manuscript.

Kind regards,

Weili Duan, Ph.D

Academic Editor

PLOS ONE

Reviewers' comments:

Reviewer's Responses to Questions

**Comments to the Author**

1. If the authors have adequately addressed your comments raised in a previous round of review and you feel that this manuscript is now acceptable for publication, you may indicate that here to bypass the “Comments to the Author” section, enter your conflict of interest statement in the “Confidential to Editor” section, and submit your "Accept" recommendation.

Reviewer #1: All comments have been addressed

Reviewer #2: (No Response)

2. Is the manuscript technically sound, and do the data support the conclusions?

Reviewer #1: Partly

Reviewer #2: Partly

3. Has the statistical analysis been performed appropriately and rigorously? 

Reviewer #1: Yes

Reviewer #2: Yes

4. Have the authors made all data underlying the findings in their manuscript fully available?

Reviewer #1: No

Reviewer #2: Yes

5. Is the manuscript presented in an intelligible fashion and written in standard English?

Reviewer #1: No

Reviewer #2: Yes

6. Review Comments to the Author

Reviewer #1: Comments have been addressed, however the manuscript is still not concise enough for a journal-standard. Authors should flash out better why the paper is important and what contribution this is bringing.

Reviewer #2: Major Items:

Thanks to the authors for revising their submission in response to reviewers’ comments on the earlier version of the paper.

Overall, and given the responses to reviewers’ comments, I wonder if it would have been much clearer to explicitly frame this paper in terms of the four basic ecosystems services (provisioning, regulating, supporting and socio-cultural) if the intention was to deploy a quant-qual methodology to show the multiple urban ecosystem services values of the Sumida Farm landscape. A table showing how these four ecosystems services are manifested in this specific case would be useful. Having abandoning the use of “resilience” as a framing concept in favour of urban ecosystems services such clarity in presentation becomes even more urgent.

I note a few additional conceptual, theoretical and epistemological challenges in my comments below.

Lines 162, 314, 568, etc. reference is made to “public trust” doctrine….the implication here is that the state is not necessarily exercising its public trust obligations in surface and ground water resources, but not enough is said about what the state actually does do in order to assess this statement. For example, if the state manages some form of licencing and regulatory oversight over water abstraction, even if wholly productionist, then it could argue that these actions discharge its public trust obligations as it understands them. A difficult arises if you, in the course of your analysis, want to argue that other water or land values related to the public trust are not being respected – but then you have to be very clear about what these may be and why they are deserving of inclusion, who says so and under what conditions? The issue of “public trust” (and its putative abrogation) would be much more compelling if there were community voices challenging the hegemonic framing of public trust in groundwater management, but this does not seem to be present. Either this issue needs to be developed so that is it clearly germane to the analysis presented, or it should be removed entirely.

L307 you say that the watercress farm “was of great value to the Sumida family and surrounding community.” – you are going to have to show how this works in the current analysis, or what you will have is a study in two parts: part 1 looking at the quantitative relations between output and a number of environmental variables on the one hand, and in part 2 a situating of the farm within Sumida family history since the 1930s and earlier favourable mention of the site and its springs in the Hawaiian language newspapers of the 19th and early 20th centuries. To really make the point about the multiple urban ecosystems values attaching to the farm, specifically, you would presumably need some sort of data from the contemporary community expressing a positive valorisation of the farm for a number of non-productivist related reasons. For example, in Line 158 you say “School and other community groups visit the farm and hear a historically common, yet currently rare story of multi-generational farming and the links between spring water and food systems.” This is good, but you need to go further and show that this story is somehow linked to broader, presumably socio-cultural, valorisations of the farm and its activities. Given the Hawaiian context, this would be easier if the crop was somehow traditional or indigenous. Do you have quotes or other data from non-Sumida sources showing that the farm is thus valued within the broader community?

Linked to the above, in L645 you say “the Sumidas’ role in the co-production of knowledge described in the current study allows them to draw on data from different fields to better understand how their crop yields can be optimized going forward.” But I am not sure that I see “co-production” emerging out of the qualitative dataset, which seems to be comprised of semi-structured interviews with the Sumidas and a study of Hawaiian language newspapers. “Co-production” is usually invoked when there are multiple “knowledges” applying to the same land (e.g. productivist and non-productivist) and there is some process of reconciling them through negotiation or brokering. Multiple knowledges/values in and of themselves are not enough to establish “co-production”. Your reframed third research question (noted in your response to reviewers) “From the perspective of the farming family, and current and historical narratives, what additional benefits are associated with Sumida Farm and Kalauao Spring that may also influence the farm’s persistence over time?” may not be, by itself, enough to establish “co-production” of knowledge.

Minor Item:

L293 “were an intentional repository” should probably be “now constitute a useful repository” UNLESS you can prove that there was a specific intention to document behind these Hawaiian language newspapers all along. The fact that they now serve this function does not prove their original intention.

7. PLOS authors have the option to publish the peer review history of their article (what does this mean?). If published, this will include your full peer review and any attached files.

Reviewer #1: No

Reviewer #2: No

---

## [Author Response · Author response to Decision Letter 1]

31 May 2020

Jennifer Engels

Hawai‘i Institute of Geophysics and Planetology

University of Hawai‘i at Mānoa

1680 East-West Road

Honolulu, HI 96822

engels@hawaii.edu

April 3, 2020

Dear Dr. Duan:

We thank you and the two reviewers for your constructive and helpful feedback to our manuscript, now entitled “Collaborative research to support urban agriculture in the face of change: the case of the Sumida watercress farm on Oʻahu” by Jennifer Engels, Sheree Watson, Henrietta Dulai, Kimberly Burnett, Christopher Wada, ‘Ano‘ilani Aga, Nathan DeMaagd, John McHugh, Barbara Sumida, and Leah Bremer. We have carefully considered each recommendation and have substantially improved the paper. The revised manuscript is now a compelling case study of the supports and challenges for urban agriculture in a changing environment. 

Below are the editor and reviewer comments in (parentheses) and our responses.

(Thank you for submitting your manuscript to PLOS ONE. After careful consideration, we feel that it has merit but does not fully meet PLOS ONE’s publication criteria as it currently stands. Therefore, we invite you to submit a revised version of the manuscript that addresses the points raised during the review process.

Please carefully deal with the comments from the second reviewer.)

We thank the editors and reviewers for the helpful comments. We have carefully considered each comment and revised the manuscript accordingly. We now feel this is well suited for publication in PLOS ONE as part of the urban ecosystems special issue.

(We would appreciate receiving your revised manuscript by April 3, 2020. )

Please see our updated financial disclosure statement in the last paragraph of the new cover letter.

(To enhance the reproducibility of your results, we recommend that if applicable you deposit your laboratory protocols in protocols.io, where a protocol can be assigned its own identifier (DOI) such that it can be cited independently in the future. For instructions see: http://journals.plos.org/plosone/s/submission-guidelines#loc-laboratory-protocols)

Our laboratory protocols are not novel. They are all previously published, and we referenced the kits and the companies where the kits were purchased which include standard protocols/instructions in the Methods descriptions. Supplementary information Tables S1, and Tables S2 provide data needed to replicate qPCR methods, standards data to assure efficiencies reported in the manuscript, and the geochemistry data as reported in the manuscript.

(Please include the following items when submitting your revised manuscript:

• A rebuttal letter that responds to each point raised by the academic editor and reviewer(s). This letter should be uploaded as separate file and labeled 'Response to Reviewers'.

• A marked-up copy of your manuscript that highlights changes made to the original version. This file should be uploaded as separate file and labeled 'Revised Manuscript with Track Changes'.

• An unmarked version of your revised paper without tracked changes. This file should be uploaded as separate file and labeled 'Manuscript'.)

Done

(Please note while forming your response, if your article is accepted, you may have the opportunity to make the peer review history publicly available. The record will include editor decision letters (with reviews) and your responses to reviewer comments. If eligible, we will contact you to opt in or out.

We look forward to receiving your revised manuscript.

Kind regards,

Weili Duan, Ph.D

Academic Editor

PLOS ONE

Reviewers' comments:

Reviewer #1: Comments have been addressed, however the manuscript is still not concise enough for a journal-standard. Authors should flash out better why the paper is important and what contribution this is bringing.)

Response: We adjusted the framing of the paper based on Reviewer 2’s comments (please see below), which also addresses Reviewer 1’s suggestion to better describe the contribution of the paper. To address a concern raised by Reviewer 2 we added an analysis of local and national press showing the community value of Sumida farm, but nonetheless managed to reduce the total word count by 1000 words to make the paper more concise. 

(Reviewer #2: Major Items:

Thanks to the authors for revising their submission in response to reviewers’ comments on the earlier version of the paper.

Overall, and given the responses to reviewers’ comments, I wonder if it would have been much clearer to explicitly frame this paper in terms of the four basic ecosystems services (provisioning, regulating, supporting and socio-cultural) if the intention was to deploy a quant-qual methodology to show the multiple urban ecosystem services values of the Sumida Farm landscape. A table showing how these four ecosystems services are manifested in this specific case would be useful. Having abandoning the use of “resilience” as a framing concept in favour of urban ecosystems services such clarity in presentation becomes even more urgent.)

Response: Thank you for this suggestion. We adjusted the framing to focus both on historical challenges, as well as the current and future multiple benefits provided by the farm. We specifically focus on crop yield (provisioning), nutrient retention and cycling (regulating and supporting), and multiple social and cultural services. We reframed the abstract, introduction, Table 1, discussion and conclusions, and the manuscript sub-headings to reflect this theme, which we agree is a compelling way to slightly shift our framing. 

We also reframed question 3 to focus specifically on the additional socio-cultural benefits provided by the farm. 

3. What additional socio-cultural benefits are provided by the farm to the Sumida family and surrounding community? 

(I note a few additional conceptual, theoretical and epistemological challenges in my comments below.

Lines 162, 314, 568, etc. reference is made to “public trust” doctrine….the implication here is that the state is not necessarily exercising its public trust obligations in surface and ground water resources, but not enough is said about what the state actually does do in order to assess this statement. For example, if the state manages some form of licencing and regulatory oversight over water abstraction, even if wholly productionist, then it could argue that these actions discharge its public trust obligations as it understands them. A difficult arises if you, in the course of your analysis, want to argue that other water or land values related to the public trust are not being respected – but then you have to be very clear about what these may be and why they are deserving of inclusion, who says so and under what conditions? The issue of “public trust” (and its putative abrogation) would be much more compelling if there were community voices challenging the hegemonic framing of public trust in groundwater management, but this does not seem to be present. Either this issue needs to be developed so that is it clearly germane to the analysis presented, or it should be removed entirely.)

Response: Thank you for pointing this out. The public trust doctrine is a commonly discussed policy related to water management in Hawai’i, but we recognize the reviewer’s point that this is distracting and complex for a beyond-Hawaiʻi audience. We removed this discussion throughout the paper to better focus on historical challenges as well as multiple current and future benefits the farm provides.

(L307 you say that the watercress farm “was of great value to the Sumida family and surrounding community.” – you are going to have to show how this works in the current analysis, or what you will have is a study in two parts: part 1 looking at the quantitative relations between output and a number of environmental variables on the one hand, and in part 2 a situating of the farm within Sumida family history since the 1930s and earlier favourable mention of the site and its springs in the Hawaiian language newspapers of the 19th and early 20th centuries. To really make the point about the multiple urban ecosystems values attaching to the farm, specifically, you would presumably need some sort of data from the contemporary community expressing a positive valorisation of the farm for a number of non-productivist related reasons. For example, in Line 158 you say “School and other community groups visit the farm and hear a historically common, yet currently rare story of multi-generational farming and the links between spring water and food systems.” This is good, but you need to go further and show that this story is somehow linked to broader, presumably socio-cultural, valorisations of the farm and its activities. Given the Hawaiian context, this would be easier if the crop was somehow traditional or indigenous. Do you have quotes or other data from non-Sumida sources showing that the farm is thus valued within the broader community?)

Response: We agree and now have drawn on articles from community newspapers and national press that point to the value of Sumida farm, including for local food production, heritage value, and contribution to local culture. While watercress is not an indigenous crop, it has a long tradition in Hawaiʻi, and is very much valued locally in Hawaiʻi by a diversity of ethnic groups. In this way, while not introduced by Polynesians, watercress is a traditional crop in the sense that it has been cultivated in Hawaiʻi for nearly a century, prior to large plantations. 

(Linked to the above, in L645 you say “the Sumidas’ role in the co-production of knowledge described in the current study allows them to draw on data from different fields to better understand how their crop yields can be optimized going forward.” But I am not sure that I see “co-production” emerging out of the qualitative dataset, which seems to be comprised of semi-structured interviews with the Sumidas and a study of Hawaiian language newspapers. “Co-production” is usually invoked when there are multiple “knowledges” applying to the same land (e.g. productivist and non-productivist) and there is some process of reconciling them through negotiation or brokering. Multiple knowledges/values in and of themselves are not enough to establish “co-production”. Your reframed third research question (noted in your response to reviewers) “From the perspective of the farming family, and current and historical narratives, what additional benefits are associated with Sumida Farm and Kalauao Spring that may also influence the farm’s persistence over time?” may not be, by itself, enough to establish “co-production” of knowledge.)

Response: While we see our work as co-production as we worked with the farm to create research questions and methodologies useful to the farm itself, we acknowledge the diversity of ways co-production can be conceptualized. Thus, we have removed co-production, and now discuss our work as a collaborative partnership. 

(Minor Item:

L293 “were an intentional repository” should probably be “now constitute a useful repository” UNLESS you can prove that there was a specific intention to document behind these Hawaiian language newspapers all along. The fact that they now serve this function does not prove their original intention.)

Response: We changed this to useful repository, although there is some evidence that this was an intentional documentation of stories and events for future generations. 

The extensive rewrites we have undertaken of the original and revised manuscripts have streamlined, focused, and clarified the objectives and outcomes of our study of the Sumida Farm/Kalauao Spring area. At this point, we believe it will make an excellent contribution to the special issue Urban Ecosystems in PLOS ONE.

Thank you very much for your consideration.

Sincerely,

Jennifer Engels

Affiliate Faculty, Hawai‘i Institute of Geophysics and Planetology

---

## [Editor Report · Decision Letter 2]

22 Jun 2020

Collaborative research to support urban agriculture in the face of change: the case of the Sumida watercress farm on O‘ahu

PONE-D-19-21079R2

Dear Dr. Jennifer,

We’re pleased to inform you that your manuscript has been judged scientifically suitable for publication and will be formally accepted for publication once it meets all outstanding technical requirements.

Kind regards,

Weili Duan, Ph.D

Academic Editor

PLOS ONE

Additional Editor Comments (optional):

The author have spent lots of time to substantially improve the paper according to the all comments from the Reviewer 2. Therefore, I think it could be accepted now.
---

## [Editor Report · Acceptance letter]

1 Jul 2020

PONE-D-19-21079R2 

Collaborative research to support urban agriculture in the face of change: the case of the Sumida watercress farm on O‘ahu 

Dear Dr. Engels:

I'm pleased to inform you that your manuscript has been deemed suitable for publication in PLOS ONE. Congratulations! Your manuscript is now with our production department. 

Kind regards, 

on behalf of

Dr. Weili Duan 

Academic Editor

PLOS ONE